# Assessing the performance of Moghani crossbred lambs derived from different mating systems with Texel and Booroola sheep

Reza Talebi[1]*, Mohsen Mardi[1], Mehrshad Zeinalabedini[1], Mehrbano Kazemi Alamouti[1], Stéphane Fabre[2], Mohammad Reza Ghaffari[1]*

1 Department of Systems and Synthetic Biology, Agricultural Biotechnology Research Institute of Iran, Agricultural Research, Education and Extension Organization (AREEO), Karaj, Iran, 2 GenPhySE, INRAE, ENVT, Université de Toulouse, Castanet Tolosan, France

* talere1986@gmail.com (RT); mrghaffari52@gmail.com (MRG)

## Abstract

In our ongoing project, which focuses on the introgression of *Booroola/FecB* gene and the *myostatin* (*MSTN*) gene into purebred Moghani sheep, we assessed the performance of second-generation Moghani crossbreds such as second crossbreds (F2) and initial back-cross generation (BC1). These crossbreds were generated through different mating systems, including in-breeding, outcrossing, first paternal backcrossing (PBC1), and first maternal backcrossing (MBC1). Notably, F2 strains exhibited lean tail, woolly fleece and a higher percentage of white coat color compared to BC1. The impact of mating systems and birth types on pre-weaning survival rates was found to be statistically significant ($P < 0.0001$), with singleton offspring resulting from paternal backcross showing a particularly substantial effect. The F2 crossbred lambs carrying the *Booroola* gene did not show a statistically significant difference in survivability compared to those carrying the *MSTN* gene, implying the *Booroola* prolificacy gene had no significant impact on survival outcomes. However, the occurrence of multiple births had a significant negative impact on lamb survival ($P < 0.0001$). The PBC1 sheep strains, specifically Texel Tamlet ram strains carrying the *MSTN* mutation, exhibited superior growth rates compared to others ($P < 0.05$). Interestingly, the *MSTN* mutation in the homozygous variant genotype significantly impacts growth rate before weaning compared to other genotypes and pure Moghani sheep ($P < 0.05$). In conclusion, this study objectively underscores the pivotal role of genetic factors, specifically through strategic mating systems like paternal backcrossing, in enhancing desired traits and growth rates in Moghani sheep, thereby contributing valuable insights to the field of sheep breeding programs.

## Introduction

Mating systems play a crucial role in influencing the performance and genetic traits of live-stock. Researchers have investigated a range of mating systems, including pure-breeding,

**Data Availability Statement:** All relevant data are within the manuscript.

**Funding:** The author(s) received no specific funding for this work.

**Competing interests:** The authors have declared that no competing interests exist.

crossbreeding, outcrossing, and backcrossing, to evaluate their effects on vital characteristics such as growth traits, lamb survival, and ewe prolificacy [1–6]. Crossbreeding with major gene carriers, using local or exotic germplasm, and genomic introgression offer promising pathways to achieve substantial genetic gains [7]. Successful sheep crossbreeding programs often involve various mating systems, such as F1 crosses, three-way crosses, and composite breeds [8]. This approach accelerates genetic progress, overcoming challenges associated with direct selection for quantitative traits, leading to more productive and profitable sheep farming operations.

Mature body size in sheep is known to be influenced by a higher degree of polygenic factors when compared to other domesticated species [9]. This means that many genes, each with a small effect, contribute to mature body size. This complex genetic nature presents a significant challenge, rendering classical breeding methods alone insufficient in achieving desired outcomes [10].

Introduction of breeds carrying major genes enables substantial improvements in muscle hypertrophy and prolificacy traits. Efforts to enhance these attributes involve crossbreeding initiatives, aiming to introduce major genetic factors. The *myostatin* (*MSTN* g+6223G>A) gene located in OAR2 region plays a key role in double muscling across different sheep breeds [11]. These breeds encompass New Zealand Texel [12, 13], Australian Texel [14], Belgian Texel [15, 16], Norwegian White Sheep [17], and the commercially relevant Charollais sheep [18]. For instance, the Texel sheep breed's *MSTN* g+6223G>A has been successfully introduced into Ramlıç sheep breeds [19, 20]. Similarly, the highly prolific *Booroola*/*FecB* allele of the *bone morphogenetic protein receptor type 1B* (*BMPR1B*) gene, initially identified in the Booroola Merino breed, has been effectively incorporated into diverse sheep breeds [8]. The *FecB* allele has resulted in sheep with increased litter size and prolificacy. Noteworthy instances include its successful integration into Afshari breed [21, 22], Assaf [23, 24], Avassi [23–25], Deccani [26], Mérinos d'Arles [27], Moghani [28, 29], and Rambouillet [30]. These examples illustrate the potential of major genes introgression to improve the genetic merit of sheep breeds.

The Moghani sheep is a local breed in Iran that is raised primarily for meat production. It is known for its large fat tail and lower prolificacy, but it also has the advantage of being able to reproduce out of season [4]. However, the Moghani sheep faces challenges in terms of economic profitability. Numerous studies have shown that the genetic progress for growth and reproductive traits in this breed is slow, due to low heritability estimates [31–34].

Herein, we launched an introgression project to enhance the productivity of Moghani sheep through strategic crossbreeding with high-yielding sheep breeds, specifically Texel and Booroola sheep. These breeds possess crucial genetic factors known for enhancing traits such as muscularity and prolificacy. The comprehensive introductory details and outcomes of the F1 crosses, including Booroola Merino×Moghani (BMM), Booroola Romney×Moghani (BRM), Texel Tamlet×Moghani (TTM), and Texel Dalzell×Moghani (TDM), have been extensively documented in our prior publications [4, 6]. In our ongoing project, we bred second-generation Moghani crossbreds using various mating systems, including in-breeding, outcrossing, and backcrossing. This study specifically focuses on a comprehensive comparison of growth performance, fat-tail traits, and lamb coat colors between purebred Moghani sheep and second-generation crossbreds. Additionally, we investigated the impact of distinct genotypes of introgressed genes, particularly prolificacy *Booroola*/*FecB* and hyper-muscularity *myostatin* (*MSTN* g+6223G>A), on the performance of these second-generation Moghani crossbreds. It's important to note that, despite our meticulous examination of these traits, we unfortunately couldn't evaluate prolificacy due to the lack of available records. Acknowledging the significance of this parameter, we plan to include it in future investigations as more data becomes accessible.

## Materials and methods

### Ethics statement

The data collection formats and procedures employed in this study underwent thorough review and approval by the Animal Care and Use Committee at the Agricultural Biotechnology Research Institute (ABRII) in Karaj, Alborz, Iran. The committee granted approval for all procedures and activities involving animals, ensuring strict adherence to local guidelines. The study exclusively relied on data obtained from live sheep at the breeding facility of Jovain Agricultural & Industrial Corporation in Jovain, Razavi Khorasan, Iran. It is important to note that no invasive procedures were conducted, and the animals were closely monitored by researchers. The study did not involve anesthesia, euthanasia, or animal sacrifice.

### Management of housing conditions, feeding regimens, and health monitoring

The animals were raised at the breeding facility of Jovain Agricultural & Industrial Corporation in Jovain, Razavi Khorasan, Iran (Jovain, Razavi Khorasan, Iran. Latitude: 36.655297/N 36˚ 39' 19.06800″, Longitude: 57.423406/57˚ 25' 24.26100″). Jovain County experiences an average annual rainfall of approximately 272 mm, with a mean daily temperature range of 17.8 to 29.5˚C, characterizing it as a moderately warm climate zone. A semi-intensive management system, characterized by a moderate amount of production inputs, was employed for animal care. The animals were permitted to graze or browse on natural pasture for approximately six hours during the daytime. Additionally, they received a supplementary diet of 0.10 to 0.40 kg concentrate mixture per day, consisting of alfalfa barn, maize silage, and salt. The amount varied based on factors such as age, physiology, and sex. Housing arrangements were organized according to sex, physiological status, and health status. Animals had access to water *ad libitum* and were subjected to vaccinations against prevalent diseases in the area. Regular treatments, deworming, and scheduled spraying were conducted to maintain their health. Each kid was assigned a unique identifying number, and their birth weight was recorded within 24 hours of birth. Kids were kept indoors with their dams for three to seven days, after which dams were moved outdoors, and kids were allowed to suckle three times a day until reaching the weaning age of 90 days. Animal care staff performed routine health assessments to ensure the overall well-being of the animals.

### Crossbreeding to produce first generation progenies

As previously detailed in [4], a total of 380 Moghani pure sheep (3-year-old ewes) underwent artificial insemination in September 2019. Frozen sperm from two New Zealand Booroola rams (one from the Merino strain and one from the Romney strain), both homozygous carriers (GG) of the *Booroola/FecB* mutation (OAR6:g.29382188A>G; NC_019463.1, Oar_v3.1, rs418841713), and from two New Zealand Texel rams (one from the Dalzell strain and one from the Tamlet strain), both homozygous carriers (AA) of the *MSTN* g+6223G>A mutation (OAR2:g.118150665G>A; NC_019459.1, Oar_v3.1, rs408469734), were used in the insemination process. The performance of the first generation of crossbred lambs (F1 crosses), including Booroola Merino×Moghani (BMM), Booroola Romney×Moghani (BRM), Texel Tamlet×Moghani (TTM), and Texel Dalzell×Moghani (TDM), was described in [4].

### Mating systems to produce second generation progenies

To produce second-generation crossbreds including second crossbreds (F2) and initial backcross generation (BC1), we used mating systems including in-breeding, outcrossing, backcrossing, and pure-breeding. The design of the mating systems is shown in Fig 1. All systems

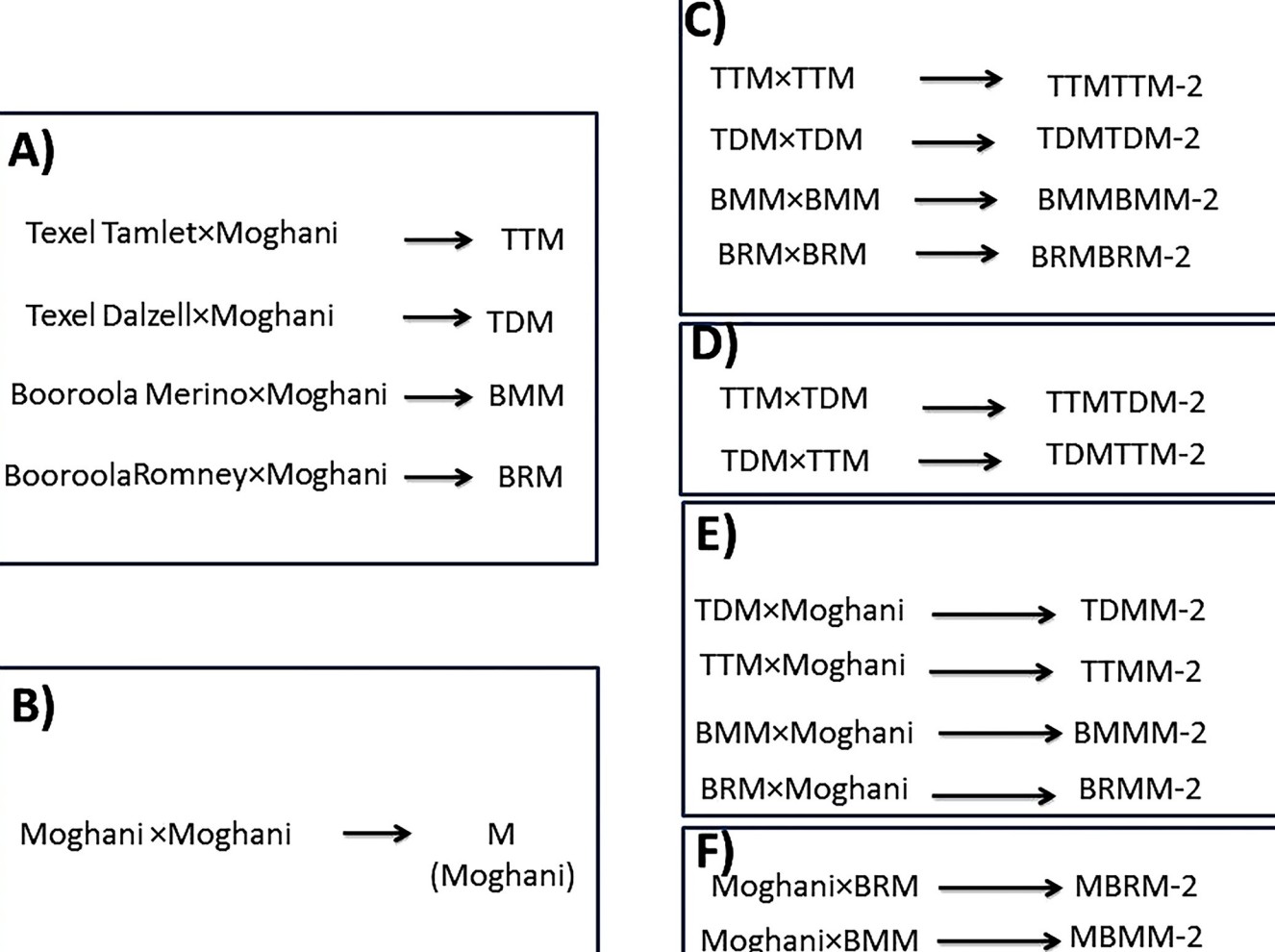

**Fig 1. Mating systems utilized to produce crossbred lambs.** (A) It depicts F1 crosses including Texel Tamlet × Moghani (TTM), Texel Dalzell × Moghani (TDM), Booroola Merino × Moghani (BMM) and Booroola Romney × Moghani (BRM). (B) It depicts pure-breeding system to produce purebred Moghani (M) lambs. (C) It depicts in-breeding system to produce inbred F2 lambs from half-sibs, Texel strains (TTMTTM-2, TDMTDM-2) and Booroola strains (BMMBMM-2, BRMBRM-2). (D) It depicts outcrossing system to produce F2 lambs from non-relative parents, Texel strains (TTMTDM-2, TDMTTM-2). (E) It depicts paternal backcrossing system to produce first generation of paternal backcross (PBC1) lambs, Texel strains (TDMM-2, TTMM-2) and Booroola strains (BMMM-2, BRMM-2). (F) It depicts maternal backcrossing system to produce first generation of maternal backcross (MBC1) lambs, Booroola strains (MBRM-2, MBMM-2).

were carried out using synchronized ewes with controlled internal drug release (CIDR) devices. In the in-breeding process, the F1 crosses were mated in half-sib states, e.g., TTM×TTM (TTMTTM-2), TDM×TDM (TDMTDM-2), BRM×BRM (BRMBRM-2) and BMM×BMM (BMMBMM-2). Therefore, the progenies had at least 12.5% inbreeding coefficient. In the out-crossing system, various strains of F1 crosses were mated while the sire and dam were not related, e.g., TDM×TTM (TDMTTM-2), TTM×TDM (TTMTDM-2). The initial paternal back-cross generation (PBC1) was established between four types of F1 crossbred rams and purebred Moghani ewes: BMM×Moghani (BMMM-2), BRM×Moghani (BRMM-2), TDM×Moghani (TDMM-2), and TTM×Moghani (TTMM-2). The initial maternal backcross generation (MBC1) was established between purebred Moghani rams (3 years old) and two types of F1 crossbred Booroola ewes: Moghani×BRM (MBRM-2) and Moghani×BMM (MBMM-2).

## Lambing information

In the context of in-breeding, a total of 25 lambs were born alive within the first week of birth, derived from a group of 39 lambs born to 25 uniparous F1 ewes aged 1 year. This group encompassed 14 F2 Booroola lambs (BRMBRM-2 and BMMBMM-2) as well as 15 F2 Texel lambs (TTMTTM-2 and TDMTDM-2). Turning to the subject of outcrossing, 40 live lambs were delivered among 48 offspring from 31 multiparous F1 ewes aged 2 years. Among these were F2 Texel lambs, exemplified by TDMTTM-2 and TTMTDM-2. In the case of PBC1, there was a count of 435 live lambs born from a pool of 438 lambs, all from 385 multiparous purebred Moghani ewes aged 3 years. The lambs were identified as TDMM-2, TTMM-2, BMMM-2, and BRMM-2. Shifting focus to MBC1, 34 live lambs were welcomed into the world among 36 offspring from 28 multiparous BMM and BRM ewes aged 2 years, showcasing the presence of MBRM-2 and MBMM-2 lambs.

## Records and data management

The growth performance of purebred Moghani, and second (F2 and BC1) generations of crossbred lambs were evaluated. Before the analysis, we adjusted weights to corresponding 90 (3-months age), and 180 (6-months age) days respectively representing adjusted 3-months weight ($W3_{adj}$) and adjusted 6-months weight ($W6_{adj}$), using the following formulas:

$$W3adj = \frac{90(W3 - BW)}{D1} + BW$$

$$Diff1 = W3adj - BW$$

$$W6adj = \frac{90(W6 - W3)}{D2} + W3adj$$

$$Diff2 = W6adj - W3adj$$

$$ADWG1 = \frac{Diff1}{90} \times 1000$$

$$ADWG2 = \frac{Diff2}{90} \times 1000$$

Where, BW = birth weight (kg), W3 = weight at 3 months of age (kg), W6 = weight at 6 months of age (kg), D1 = number of days between birth day and the 3-months weighing, D2 = number of days between the 3-month and 6-months weighings, Diff1 = difference of weights from birth to 3 months (kg), Diff2 = difference of weights from 3 to 6 months (kg), ADWG1 = average daily weight gain from birth to 3 months (gr), and ADWG2 = average daily weight gain from 3 to 6 months (gr). Additionally, fat-tail traits at 6 months of age were documented, encompassing tail/fat tail type, fat-tail height (FTH, in centimeters), and fat-tail width (FTW, in centimeters). Furthermore, morphometric of the lamb coat colors were collected in five coat colors types including white, brown, light-brown, strong brown, and black.

## Extraction of DNA and genotyping through PCR-RFLP method

Blood sampling was collected two to three months after birth. Genomic DNA was extracted using the procedure closely followed the methodology as previously described [35]. The PCR-RFLP genotyping procedures for *BMPR1B* mutation OAR6:g.29382188A>G and the

*MSTN* mutation OAR2:g.118150665G>A were conducted following the methods outlined in [4].

## Statistical analysis

The frequency of colors, mating systems, sex, introgressed gene, and type of birth were compared based on two-way chi-square tests with a significance level of 5%, using the PROC FREQ procedure in SAS Version 8.2 [36].

The analysis of growth traits utilized the general linear model (GLM) procedure implemented in SAS. The determination of significance ($P < 0.05$) was conducted through Duncan's multiple range test. The analysis employed the following multivariate models:

$$y_{ijklmnopqr} = \mu + B_j + D_k + C_l + T_m + S_n + M_o + G_p + L_q + (B \times D)_{jk} + (C \times G)_{lp}$$
$$+ (M \times G)_{op} + e_{ijklmnopqr}$$

where: $y_{ijklmnopqr}$ is the vector of observation of the $i^{th}$ animal within the $j^{th}$ sire breed, $k^{th}$ dam breed, $l^{th}$ sheep strain, $m^{th}$ type of birth, $n$ sex category, $o^{th}$ mating system, $p^{th}$ genotype category, and $q^{th}$ tail type.

$\mu$ is the overall mean, $B_j$ is the effect of the $j^{th}$ sire breed ($j$ = Moghani, TTM, TDM, BMM, BRM), $D_k$ is the effect of the $k^{th}$ dam breed ($k$ = Moghani, TTM, TDM, BMM, BRM), $C_l$ is the effect of the $l^{th}$ sheep strain ($l$ = Moghani, BRMBRM-2, BMMBMM-2, BMMM-2, BRMM-2, MBRM-2, MBMM-2, TTMTTM-2, TDMTDM-2, TDMTTM-2, TTMTDM-2, TDMM-2, TTMM-2), $T_m$ is the effect of $m^{th}$ type of birth ($m$ = 1, 2, 3, 4), $S_n$ is the effect of $n^{th}$ sex ($n$ = male and female), $M_o$ is the effect of $o^{th}$ mating system ($o$ = in-breeding, outcrossing, paternal backcrossing, maternal backcrossing, and pure-breeding), $G_p$ is the effect of $p^{th}$ genotype (GG, GA, AA for *MSTN* mutation and AA, AG, GG for *BMPR1B* mutation), $L_q$ is the effect of $q^{th}$ tail type ($q$ = tail or fat-tailed), $(B \times D)_{jk}$ is the interaction effect between $j^{th}$ sire breed and $k^{th}$ dam, $(C \times G)_{lp}$ is the interaction effect between $l^{th}$ strain and $p^{th}$ genotype, $(M \times G)_{op}$ is the interaction effect between $o^{th}$ mating system and $p^{th}$ genotype.

## Results and discussion

This investigation builds upon our prior studies [4, 6] regarding the introgression of the prolificacy *Booroola/FecB* mutation (OAR6:g.29382188A>G; NC_019463.1, Oar_v3.1, rs418841713) and the hyper-muscularity *MSTN* g+6223G>A mutation (OAR2: g.118150665G>A; NC_019459.1, Oar_v3.1, rs408469734) into purebred Moghani sheep. We conducted a systematic comparison of growth traits, fat-tail characteristics, and morphometrics of lamb coat colors between purebred Moghani sheep and their second generations (F2 and BC1) of crossbred lambs. Additionally, we evaluated the effects of each genotype (homozygous reference, heterozygous, and homozygous variant) of the *BMPR1B* and *MSTN* mutations on the body weights, growth traits, fat-tail traits, and morphometrics of lamb coat colors in second-generation crossbred lambs

### RFLP genotyping results

The RFLP genotyping analysis of the *MSTN* mutation (OAR_v.3.1; Chr 2: g.118150665G>A) provided confirmation regarding the genotypes of F2 crossbred lambs with Texel. These genotypes included homozygous reference G/G, heterozygous G/A, and homozygous variant A/A (Fig 2A and S1 Fig). Furthermore, the RFLP genotyping of the *BMPR1B/Booroola* mutation (OAR_v.3.1; Chr 6: g.29382188A>G) revealed the genotypic variations within F2 crossbred lambs with Booroola Merino and Booroola Romney. The genotypes encompassed

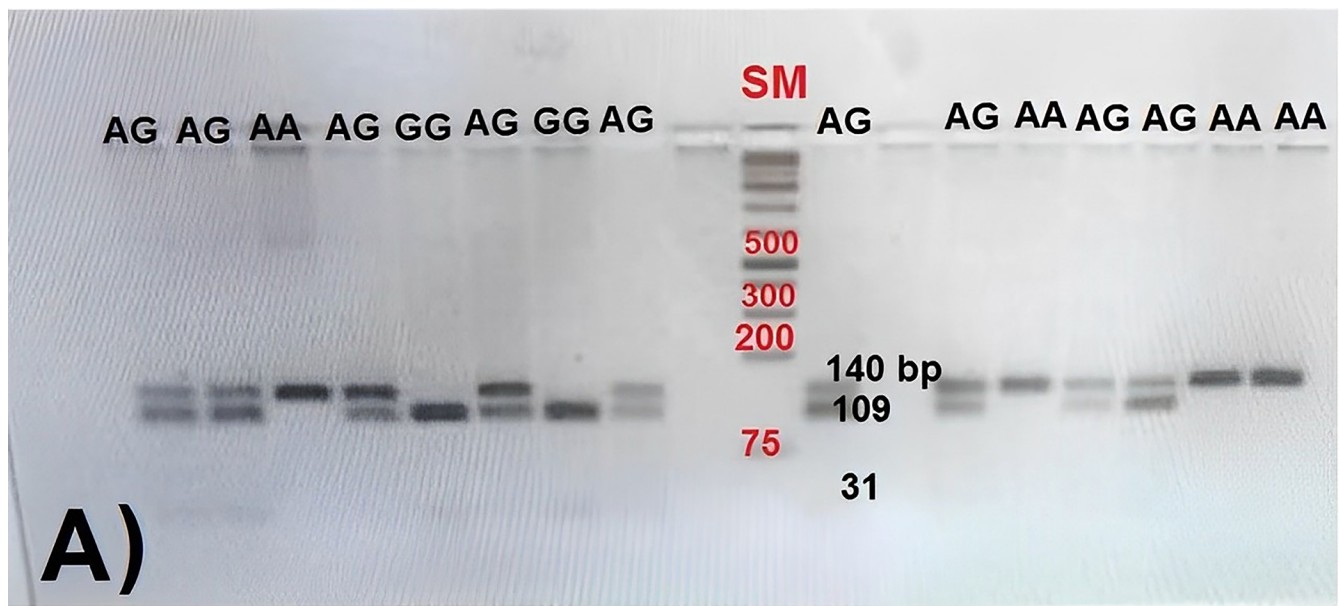

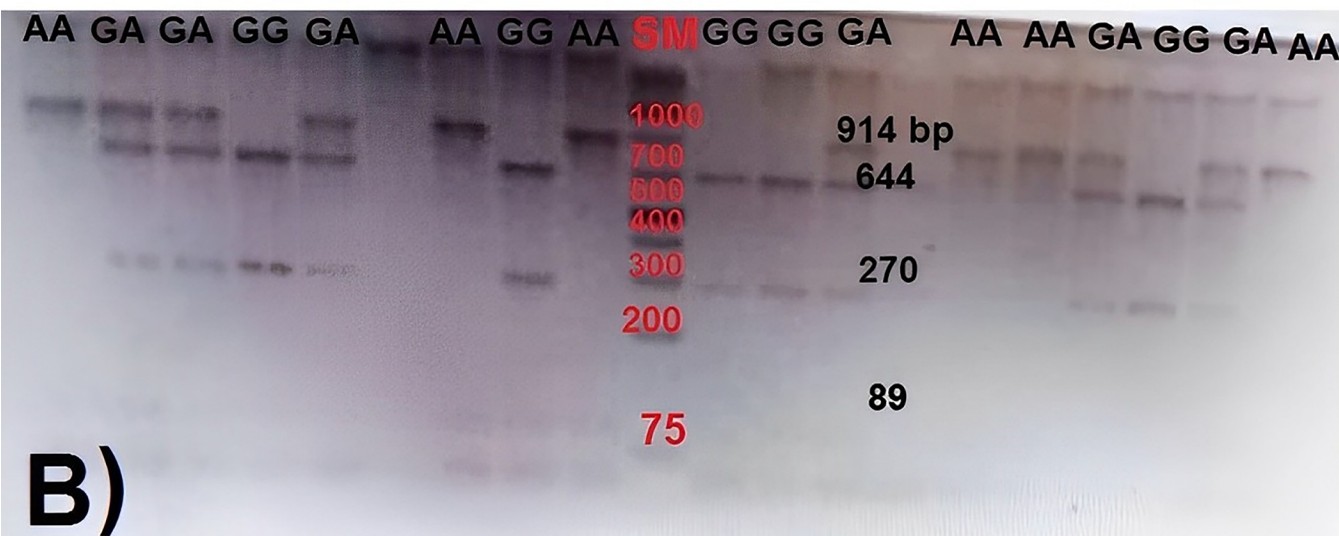

**Fig 2. The electrophoretic result of PCR-RFLP band patterns migrated on agarose gel 3.5%.** (A) PCR amplified *BMPR1B* gene digested by *Ava*II (5'-G/ GWCC) restriction enzyme for *Booroola/FecB* mutation (OAR6: 29382188A>G, NC_019463.1 of genome assembly Oar_v3.1). The lanes depict *Booroola* mutation genotypes including homozygous reference (AA), heterozygous (AG) and homozygous variant (GG). SM: GeneRuler 1 kb Plus DNA Ladder #SM1331. (B) PCR amplified ovine *myostatin* gene (*MSTN*) digested by *HpyCH*4IV (5'- A/CGT) restriction enzyme for *MSTN* mutation (OAR2: 118150665G>A, NC_019463.1 of genome assembly Oar_v3.1). The lanes depict *MSTN* mutation genotypes including homozygous reference (GG), heterozygous (GA) and homozygous variant (AA). SM: GeneRuler 1 kb Plus DNA Ladder #SM1331.

homozygous reference A/A, heterozygous A/G, and homozygous variant G/G at this specific locus (Fig 2B and S1 Fig). It is noteworthy that lambs with the homozygous variant genotype for both genes have exclusively been discovered in progenies resulting from a combination of in-breeding and outcrossing systems. A study by Daetwyler et al. [37] revealed a negative link between inbreeding rates and heritability. This is due to reduced genetic diversity caused by inbreeding. Hence, it underscores the necessity for precise breeding management to regulate these genetic variations.

## Phenotype and morphometric characteristics

The initial observable traits in the F1 crossbred lambs were a slender tail, a white coat, and a woolly white fleece. The purebred Moghani sheep, on the other hand, had a substantial fat-tail and a light-brown fleece that was either woolly or hairy [4]. Consistent with the results reported by Khaldari et al. [38, 39], the mating of lean-tailed rams with fat-tailed ewes resulted in the birth of F1 crossbred lambs with slender tails, regardless of gender. Moving on to the second-generation of crossbred lambs, twelve strains, including TTMTDM-2, TTMTTM-2, TDMTDM-2, TDMTTM-2, BRMBRM-2, BMMBMM-2, TDMM-2, TTMM-2, BRMM-2, BMMM-2, MBMM-2, and MBRM-2, generated through different mating systems, have been shown in Fig 3. The offspring resulting from both in-breeding and outcrossing exhibited a lean tail, and a woolly fleece (Fig 4A–4D). In the context of backcrossing progeny, these lambs displayed varying from short (including fat-rumped) to large fat-tails, as well as a woolly/hairy fleece (Fig 4E). The shape and size of a sheep's tail play a crucial role in their genetics and have important implications for their domestication, ability to thrive in various environments, productivity, and animal welfare [40]. At present, the fat-tail phenotype is not favorable for inclusion in breeding programs in Iran, as it presents several adverse consequences that are likely to impact aspects such as animal mobility, mating, food efficiency, and breeding expenses [4, 11].

As indicated in Table 1, the F2 crossbred strains, including BRMBRM-2, BMMBMM-2, TTMTTM-2, TDMTDM-2, TDMTTM-2, and TTMTDM-2, exhibited a notably higher

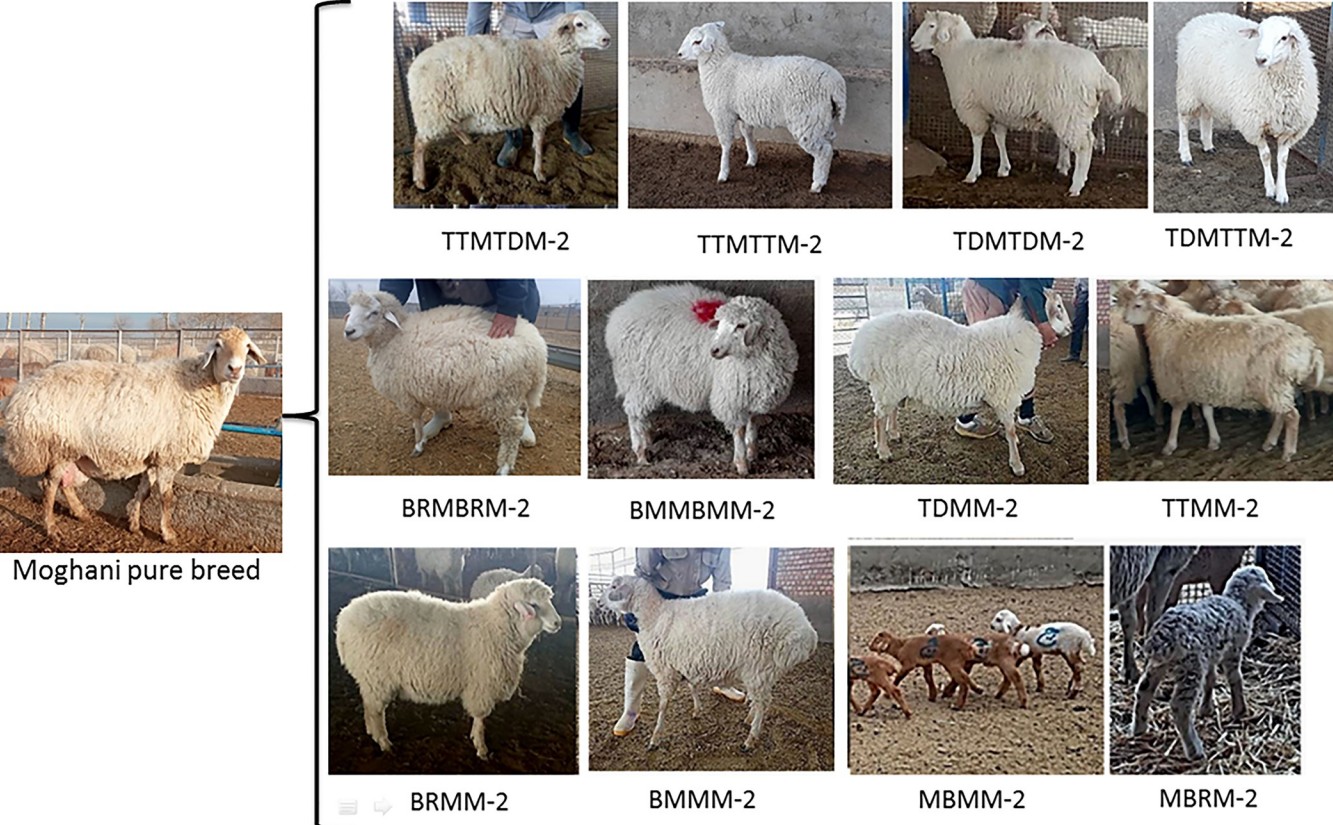

**Fig 3. Sheep strains generated through different mating systems.** Twelve second-generation crossbred lamb strains, derived from the Moghani pure breed maternal lineage, include TTMTDM-2, TTMTTM-2, TDMTDM-2, TDMTTM-2, BRMBRM-2, BMMBMM-2, TDMM-2, TTMM-2, BRMM-2, BMMM-2, MBMM-2, and MBRM-2.

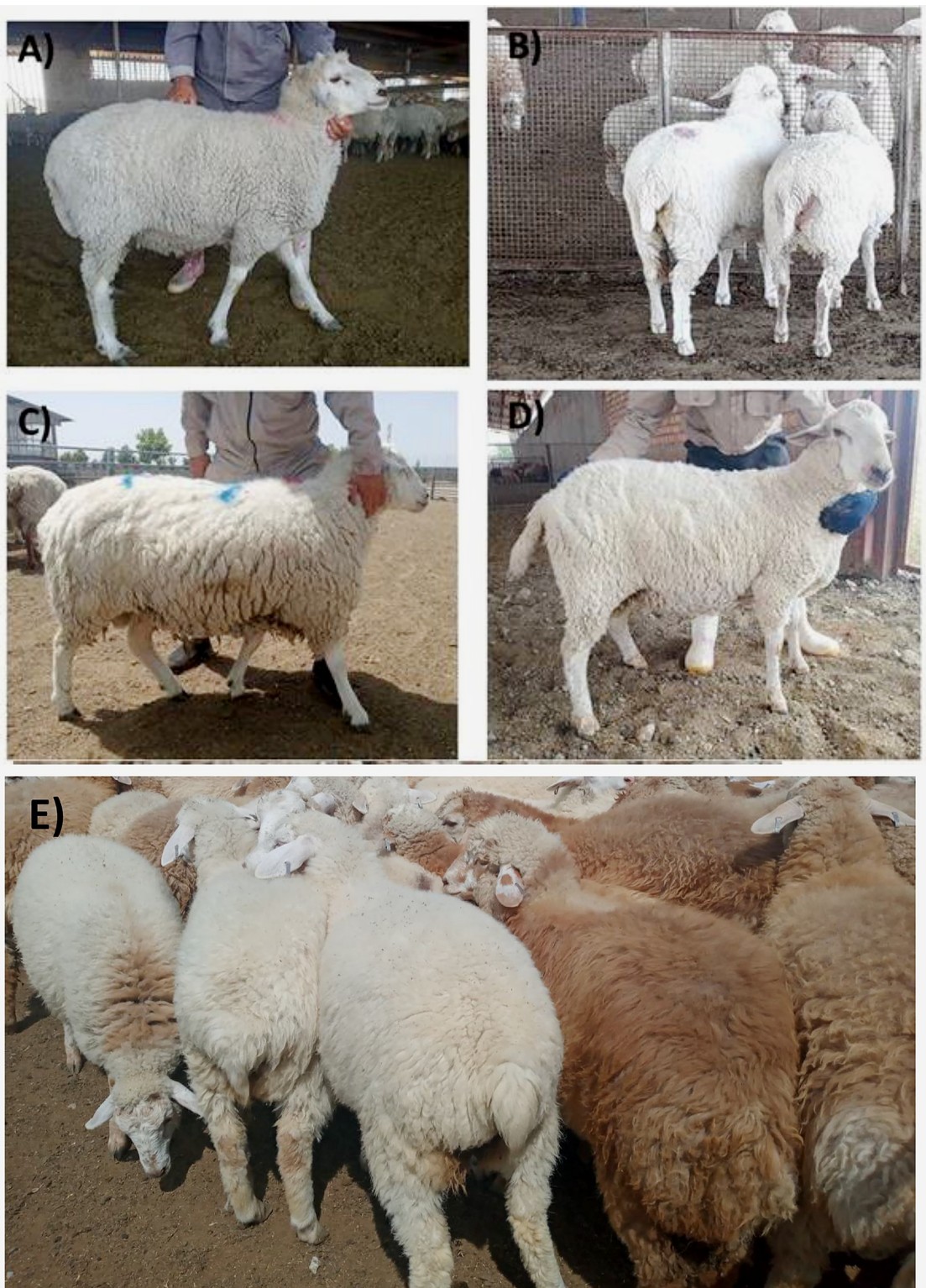

**Fig 4. Phenotype characteristic of second-generation crossbreds lambs (F2 and BC1).** (A) The F2 offspring resulting from in-breeding with the homozygous variant (GG) genotype for *Booroola/FecB* mutation (OAR6: 29382188A>G, NC_019463.1 of genome assembly Oar_v3.1). (B) The F2 offspring resulting from in-breeding with the homozygous variant genotype (AA) for *MSTN* mutation (OAR2: 118150665G>A, NC_019463.1 of genome assembly Oar_v3.1). (C) The F2 offspring resulting from outcrossing with the homozygous variant genotype (AA) for *MSTN* mutation (OAR2: 118150665G>A, NC_019463.1 of genome assembly

Oar_v3.1). (D) The F2 offspring resulting from outcrossing with the homozygous variant (GG) genotype for *Booroola/FecB* mutation (OAR6: 29382188A>G, NC_019463.1 of genome assembly Oar_v3.1). (E) The BC1 offspring resulting from initial backcross generation.

percentage of white coat color compared to the BC1 strains, which encompassed BMMM-2, BRMM-2, TDMM-2, TTMM-2, MBRM-2, and MBMM-2. Notably, all lambs born from the BMMBMM-2 strain displayed a white coat color (see Table 1). It is noteworthy that none of the parents with a black coat color were intentionally selected from the F1 crossbred and Moghani pure sheep populations to create the F2 and BC1 strains. Nonetheless, we observed the occurrence of black coat color in the sheep strains listed in Table 1, including BMMM-2 (1.02%), BRMM-2 (1.27%), MBMM-2 (5.88%), TTMTDM-2 (16.67%), TDMM-2 (2.88%), and TTMM-2 (1.52%). Given the intricate nature of the coat color trait, it has been proposed that the phenotypic variation in coat colors may be influenced by multiple genes. It was expected that various loci or genes would interact epistatically, contributing to the broad spectrum of sheep coat colors [41].

## Preweaning lamb survival

The data pertaining to preweaning lamb survival across various variables such as mating systems, sex, introgressed gene, and type of birth has been detailed in Table 2. The mating systems and type of birth significantly affected lamb's pre-weaning survival rates (refer to Table 2, $P < 0.0001$). Notably, singleton offspring resulting from paternal backcross exhibited a substantial impact. Further, females displayed a slightly improved survival rate compared to males, the observed differences did not reach statistical significance (refer to Table 2, $P > 0.05$). Consequently, it can be inferred that the survival rates of females and males are similar. These findings align with prior researches [5, 42] that also noted fluctuations in lamb survival rates. Our study corroborates these observations and indicates that the presence of introgressed *MSTN* and *Booroola* genes does not exert a noteworthy impact on lamb survivability (see Table 2). Intriguingly, F2 crossbred lambs carrying the *Booroola* gene had no

**Table 1. Percentage rate of coat color patterns across produced strains of progenies between pure Moghani sheep and second generations of crossbred lambs (F2 and BC1).**

| Breed | Strains | N | Percentage rate of coat color | | | | |
|---|---|---|---|---|---|---|---|
| | | | white | brown | light-brown | strong brown | black |
| Booroola crossbred lambs | BRMBRM-2 | 17 | 61.11 | 0.00 | 33.33 | 5.56 | 0.00 |
| | BMMBMM-2 | 4 | 100.00 | 0.00 | 0.00 | 0.00 | 0.00 |
| | BMMM-2 | 96 | 33.67 | 35.71 | 29.59 | 0.00 | 1.02 |
| | BRMM-2 | 153 | 34.18 | 37.34 | 27.22 | 0.00 | 1.27 |
| | MBRM-2 | 15 | 6.67 | 40.00 | 53.33 | 0.00 | 0.00 |
| | MBMM-2 | 17 | 0.00 | 41.18 | 52.94 | 0.00 | 5.88 |
| Texel crossbred lambs | TTMTTM-2 | 8 | 75.00 | 0.00 | 25.00 | 0.00 | 0.00 |
| | TDMTDM-2 | 9 | 77.78 | 0.00 | 11.11 | 11.11 | 0.00 |
| | TDMTTM-2 | 14 | 85.71 | 0.00 | 14.29 | 0.00 | 0.00 |
| | TTMTDM-2 | 12 | 33.33 | 16.67 | 33.33 | 0.00 | 16.67 |
| | TDMM-2 | 96 | 36.54 | 26.92 | 33.65 | 0.00 | 2.88 |
| | TTMM-2 | 63 | 46.97 | 30.30 | 21.21 | 0.00 | 1.52 |
| Pure breed | Moghani | 172 | 0.00 | 0.00 | 100.00 | 0.00 | 0.00 |

N: number of lambs at each group

**Table 2. Preweaning survival rate (%) across mating systems, sex, introgressed gene, and type of birth in second (F2 and BC1) generations of crossbred lambs.**

| Mating systems | Survival rate (%) | Chi.Square ($X^2$) | P Value | Sex | Survival rate (%) | Chi.Square ($X^2$) | P Value | Introgressed gene | Survival rate (%) | Chi.Square ($X^2$) | P Value | Type of birth | Survival rate (%) | Chi.Square ($X^2$) | P Value |
|---|---|---|---|---|---|---|---|---|---|---|---|---|---|---|---|
| in-breeding | 64.10 | 162.84 | <0.0001 | male | 93.28 | 0.261 | 0.609 | MSTN | 92.07 | 1.982 | 0.159 | singleton | 96.40 | 44.682 | <0.0001 |
| outcrossing | 57.14 | | | female | 94.33 | | | Boorroola | 95.01 | | | twins | 91.04 | | |
| paternal backcross | 99.31 | | | | | | | | | | | triplets | 33.33 | | |
| maternal backcross | 94.44 | | | | | | | | | | | quadruplets | 26.67 | | |

significant difference in survivability compared to lambs carrying the *MSTN* gene, as indicated in Table 2. The findings suggest that the *Booroola* prolificacy gene does not have a detrimental impact on lamb survivability. However, it is important to highlight that we observed a decline in lamb survival during cases of multiple births (refer to Table 2). This observation aligns with previous unfavorable findings when the *Booroola* mutation was introduced into Australian [43] and American [44] sheep breeds. In these studies, lamb mortality notably increased among highly prolific ewes managed in extensive conditions. This occurrence can be linked to the counteractive correlation between heightened prolificacy and lamb survival rates, combined with an increased susceptibility to pregnancy toxemia in ewes [8]. The ability of fetuses to resist hypoxia is critically significant in pregnancies with multiple fetuses, as hypoxia can have a significant impact on fetal survival and birth weight [45].

## Effect of birth type on growth performance of lambs

The impact of birth type on the growth performance of second generations of crossbred lambs (F2 and BC1) is presented in Table 3. Pre-weaning weights, such as birth weight (BW) and weaning weight ($W3_{adj}$), exhibited a significant difference. Quadruplets and triplet-born lambs displayed notably lower weights compared to twins and single-born lambs ($P < 0.0001$, see Table 3). In contrast, post-weaning, triplets exhibited a faster growth rate than single-born, twins, and quadruplet-born lambs, with an increase of 255.06 g/d ($P < 0.9129$, refer to Table 3). These findings align with the results of our prior study on F1 crossbreds [4]. A study has demonstrated that ewes rearing triplets produce 21% more milk and exhibit greater feed-to-milk conversion efficiency when compared to ewes of similar weight rearing twins [46]. This is likely attributable to the influence of the number of lambs nursed on ewe lactation. In contrast, McHugh et al. [47] found that lambs born and reared as triplets exhibited a notably slower growth rate of 299 g/d. Additionally, in a study examining the impact of the *Booroola* gene on the growth performance of Garole × Malpura sheep, Kumar et al. [48] found that the type of birth had a significant effect ($P < 0.01$) on body weight from birth to 12 months of age. Notably, single-born lambs exhibited a significantly higher body weight ($P < 0.01$) compared to twins and triplets within the same age range. Furthermore, the type of birth had a significant

**Table 3. Predicted means for the type of birth effects on lamb growth traits of pure Moghani sheep and second generations of crossbred lambs (F2 and BC1).**

| Traits | Type of birth | | | | | | | | SEM | *P* Value |
|---|---|---|---|---|---|---|---|---|---|---|
| | Singletons | | Twins | | Triplets | | Quadruplets | | | |
| | N | Mean | N | Mean | N | Mean | N | Mean | | |
| BW (kg) | 428 | 4.70 [a] | 237 | 4.11 [a] | 4 | 2.80 [b] | 7 | 2.67 [b] | 0.23 | <0.0001 |
| $W3_{adj}$ (kg) or weaning | 428 | 26.67 [a] | 237 | 25.20 [ab] | 2 | 23.20 [ab] | 7 | 22.55 [b] | 0.31 | <0.0001 |
| $W6_{adj}$ (kg) | 428 | 47.50 | 237 | 46.97 | 2 | 46.14 | 7 | 43.01 | 0.48 | 0.1798 |
| Diff1 (kg) | 428 | 22.13 [a] | 237 | 21.06 [ab] | 2 | 20.58 [ab] | 7 | 19.88 [b] | 0.19 | 0.0009 |
| Diff2 (kg) | 428 | 20.76 | 237 | 21.90 | 2 | 22.96 | 7 | 20.47 | 0.36 | 0.9129 |
| ADWG1 (gr) | 428 | 245.85 [a] | 237 | 234.01 [ab] | 2 | 228.71 [ab] | 7 | 220.87 [b] | 2.10 | 0.0009 |
| ADWG2 (gr) | 428 | 230.63 | 237 | 243.16 | 2 | 255.06 | 7 | 227.41 | 4.01 | 0.9129 |

N: number of lambs at each group; SEM: standard error of mean; BW: birth weight; $W3_{adj}$: adjusted weight at 3 months (weaning); $W6_{adj}$: adjusted weight at 6 months; Diff1: difference of weights at birth to 3 months; Diff2: difference of weights at 3–6 months; ADWG1; average daily weight gain from birth to 3 months; ADWG2: average daily weight gain from 3 to 6 months. a,b: The means with the same letter in each row are not significantly different by Duncan's multiple range test at 0.05 level.

**Table 4. Predicted means for the effects of sex on lamb growth traits and fat-tail measurements in second generations of crossbred lambs (F2 and BC1).**

| | | N | Lamb growth traits | | | | | | | Lamb fat-tail measurements | |
|---|---|---|---|---|---|---|---|---|---|---|---|
| | | | BW (kg) | W3$_{adj}$(kg) | W6$_{adj}$ (kg) | Diff1 (kg) | Diff2 (kg) | ADWG1 (gr) | ADWG2 (gr) | FTH (cm) | FTW (cm) |
| **Sex** | Male | 309 | 4.70 [a] | 27.50 [a] | 52.20 [a] | 22.94 [a] | 24.70 [a] | 254.90 [a] | 274.30 [a] | 15.48 [b] | 18.12 [a] |
| | Female | 367 | 4.26 [b] | 24.95 [b] | 42.94 [b] | 20.73 [b] | 18.073 [b] | 230.30 [b] | 201 [b] | 18.10 [a] | 15.53 [b] |
| | SEM | | 0.04 | 0.17 | 0.39 | 0.16 | 0.33 | 1.81 | 3.69 | 0.32 | 0.28 |
| | *P* Value | | <0.0001 | <0.0001 | <0.0001 | <0.0001 | <0.0001 | <0.0001 | <0.0001 | <0.05 | <0.0001 |

N =: number of lambs at each mating strategy; BW: birth weight; W3$_{adj}$: adjusted weight at 3 months (weaning); W6$_{adj}$: adjusted weight at 6 months; Diff1: difference of weights at birth to 3 months; Diff2: difference of weights at 3–6 months; ADWG1; average daily weight gain from birth to 3 months; ADWG2: average daily weight gain from 3 to 6 months; FTH: fat-tail height; FTW: fat-tail width. a,b: The means with the same letter in each column are not significantly different in Duncan's multiple range test at 0.05 level.

impact ($P < 0.01$) on the average daily weight gain before weaning, while it did not significantly affect the average daily weight gain after weaning.

## Effects of sex and mating systems on growth performance of lambs and fat-tail traits

In the present study the female lambs have significantly lower growth rate compared to males (Table 4). Moreover, female lambs have greater FTH (18.10 cm vs. 15.48 cm) but lower FTW (15.53 cm vs. 18.12 cm) than male lambs ($P < 0.05$).

Fig 1 illustrates the use of four distinct mating systems (Fig 1C–1F) in producing crossbred lambs of F2 and BC1. Notably, the offspring resulting from paternal backcrosses exhibited significantly greater birth weights compared to those from the other mating systems ($P < 0.0001$, Table 5). While lambs born through pure-breeding, in-breeding, and outcrossing displayed higher average daily weight gain before weaning (ADWG1) when contrasted with the backcrossed lambs, the situation shifted post-weaning. After weaning, the lambs born through backcrossing demonstrated a significantly higher growth rate (ADWG2) compared to the others ($P < 0.0001$, Table 5). To enhance the growth rate, we have determined that paternal backcrossing is the optimal strategy for the introgression of major genes into the Moghani pure breed. When it comes to fat-tail traits, lambs born from pure-breeding Moghani sheep exhibited significantly greater FTH and FTW compared to lambs born from various mating systems, including in-breeding, outcrossing, paternal backcrossing, and maternal backcrossing ($P < 0.0001$, Table 5).

## Effects of progeny strains on growth performance of lambs and fat-tail traits

In this study, strains resulting from paternal backcrossing, namely TTMM-2, TDMM-2, BMMM-2, and BRMM-2, exhibited greater birth weight (BW), as well as W3$_{adj}$ and W6$_{adj}$ measurements, and growth rates when compared to other sheep strains ($P < 0.05$, Table 6). Conversely, strains derived from maternal backcrossing systems, such as MBRM-2 and MBMM-2, displayed lower body weight compared to pure Moghani sheep before weaning. However, after weaning, they exhibited significantly higher growth rates ($P < 0.05$, Table 6).

**Table 5. Predicted means for the mating systems on lamb growth traits and fat-tail measurements of pure Moghani sheep and second generations of crossbred lambs (F2 and BC1).**

| Mating strategy | N | Lamb growth traits | | | | | | | Fat-tail measurements | |
|---|---|---|---|---|---|---|---|---|---|---|
| | | BW (kg) | W3$_{adj}$(kg) or weaning | W6$_{adj}$(kg) | Diff1 (kg) | Diff2 (kg) | ADWG1 (gr) | ADWG2 (gr) | FTH (cm) | FTW (cm) |
| **in-breeding** | 27 | 3.13 c | 24.89 $^{bc}$ | 37.77 $^d$ | 21.60 $^{ab}$ | 12.88 $^d$ | 240.00 $^{ab}$ | 143.10 $^d$ | 21.27 $^b$ | 1.57 $^d$ |
| **outcrossing** | 20 | 4.00 b | 26.18 $^{ab}$ | 43.53 $^{bc}$ | 21.94 $^{ab}$ | 17.34 $^c$ | 243.74 $^{ab}$ | 192.69 $^c$ | 16.05 $^c$ | 0.85 $^d$ |
| **paternal backcrossing** | 425 | 4.93 a | 25.98 $^{ab}$ | 50.87 $^a$ | 21.26 $^b$ | 24.94 $^a$ | 236.24 $^b$ | 277.14 $^a$ | 12.59 $^c$ | 17.11 $^b$ |
| **maternal backcrossing** | 32 | 3.40 c | 23.93 $^c$ | 44.10 $^b$ | 20.53 $^b$ | 20.16 $^b$ | 228.13 $^b$ | 224.04 $^b$ | 11.66 $^c$ | 12.69 $^c$ |
| **pure-breeding** | 172 | 3.88 b | 27.02 $^a$ | 40.24 $^{cd}$ | 23.14 $^a$ | 13.10 $^d$ | 257.08 $^a$ | 145.50 $^d$ | 28.29 $^a$ | 19.53 $^a$ |
| SEM | | 0.04 | 0.17 | 0.39 | 0.16 | 0.33 | 1.81 | 3.69 | 0.32 | 0.28 |
| P Value | | <0.0001 | <0.0001 | <0.0001 | <0.0001 | <0.0001 | <0.0001 | <0.0001 | <0.0001 | <0.0001 |

N =: number of lambs at each mating strategy; BW: birth weight; W3$_{adj}$: adjusted weight at 3 months (weaning); W6$_{adj}$: adjusted weight at 6 months; Diff1: difference of weights at birth to 3 months; Diff2: difference of weights at 3–6 months; ADWG1; average daily weight gain from birth to 3 months; ADWG2: average daily weight gain from 3 to 6 months; FTH: fat-tail height; FTW: fat-tail width. a,b: The means with the same letter in each column are not significantly different in Duncan's multiple range test at 0.05 level.

Among the various strains assessed, TTMM-2 exhibited superior characteristics, including higher BW, W3$_{adj}$, W6$_{adj}$, Diff2, and ADWG2 in comparison to other strains. However, Moghani pure sheep displayed greater Diff1 and ADWG1 than TTMM-2, although the difference was not statistically significant (refer to Table 6). TTMM-2 originated from the

**Table 6. Predicted means for the produced strains of progenies on growth traits and fat-tail measurements between pure Moghani sheep and second generations of crossbred lambs (F2 and BC1).**

| Breed | Strains | N | Lamb growth traits | | | | | | | Fat-tail measurements | |
|---|---|---|---|---|---|---|---|---|---|---|---|
| | | | BW (kg) | W3$_{adj}$ (kg) or weaning | W6$_{adj}$ (kg) | Diff1 (kg) | Diff2 (kg) | ADWG1 (gr) | ADWG2 (gr) | FTH (cm) | FTW (cm) |
| **Booroola crossbred lambs** | **BRMBRM-2** | 17 | 2.72 $^d$ | 20.95 $^e$ | 36.45 $^{ef}$ | 18.02 $^d$ | 15.49 $^{dc}$ | 200.15 $^d$ | 172.13 $^{cd}$ | 21.25 $^{bc}$ | 3.14 $^d$ |
| | **BMMBMM-2** | 4 | 2.70 $^d$ | 22.24 $^{de}$ | 31.17 $^f$ | 19.54 $^{cd}$ | 8.94 $^e$ | 217.05 $^{cd}$ | 99.3 $^e$ | 25 $^{ab}$ | 0 $^d$ |
| | **BMMM-2** | 96 | 4.93 $^a$ | 26.10 $^{abc}$ | 50.73 $^{ab}$ | 21.23 $^{abc}$ | 24.67 $^{ab}$ | 235.84 $^{bc}$ | 274.12 $^{ab}$ | 13.15 $^d$ | 17.89 $^a$ |
| | **BRMM-2** | 153 | 4.9 $^a$ | 26.40 $^{abc}$ | 52.23 $^a$ | 21.65 $^{abc}$ | 25.83 $^a$ | 240.60 $^{abc}$ | 287 $^a$ | 12.97 $^d$ | 16.38 $^{ab}$ |
| | **MBRM-2** | 15 | 3.34 $^c$ | 23.80 $^{cde}$ | 44.72 $^{bcd}$ | 20.44 $^{bcd}$ | 20.94 $^{ab}$ | 227.06 $^{bcd}$ | 232.65 $^{ab}$ | 12.47 $^d$ | 12.47 $^c$ |
| | **MBMM-2** | 17 | 3.46 $^c$ | 24.10 $^{bcde}$ | 43.55 $^{bcde}$ | 20.62 $^{bcd}$ | 19.48 $^{bc}$ | 229.06 $^{bcd}$ | 216.43 $^{bc}$ | 10.94 $^d$ | 12.88 $^{bc}$ |
| **Texel crossbred lambs** | **TTMTTM-2** | 8 | 3.75 $^{bc}$ | 27.81 $^a$ | 42.77 $^{cde}$ | 24.03 $^a$ | 14.95 $^{dc}$ | 267 $^a$ | 166.14 $^{cd}$ | 17 $^{cd}$ | 0 $^d$ |
| | **TDMTDM-2** | 9 | 3.50 $^c$ | 27.76 $^a$ | 37.95 $^{def}$ | 24.26 $^a$ | 10.2 $^{de}$ | 269.5 $^a$ | 113.31 $^{de}$ | 20.5 $^{bc}$ | 0 $^d$ |
| | **TDMTTM-2** | 14 | 4.23 $^b$ | 27.37 $^{ab}$ | 48.58 $^{abc}$ | 22.95 $^{ab}$ | 21.21 $^{ab}$ | 255 $^{ab}$ | 235.6 $^{ab}$ | 16.64 $^{cd}$ | 1.55 $^d$ |
| | **TTMTDM-2** | 12 | 3.72 $^{bc}$ | 24.73 $^{abcd}$ | 37.35 $^{ef}$ | 20.7 $^{bcd}$ | 12.62 $^{de}$ | 229.98 $^{bcd}$ | 140.24 $^{de}$ | 15.33 $^{cd}$ | 0 $^d$ |
| | **TDMM-2** | 96 | 4.80 $^a$ | 24.52 $^{abcd}$ | 47.69 $^{abc}$ | 20.10 $^{bcd}$ | 23.39 $^{ab}$ | 223.38 $^{bcd}$ | 259.88 $^{ab}$ | 11.59 $^d$ | 16.57 $^{ab}$ |
| | **TTMM-2** | 63 | 5.22 $^a$ | 27.16 $^{abc}$ | 52.82 $^a$ | 22.20 $^{abc}$ | 25.67 $^a$ | 246.45 $^{abc}$ | 285.2 $^a$ | 12.39 $^d$ | 18.58 $^a$ |
| **Pure breed** | **Moghani** | 172 | 3.88 $^{bc}$ | 27.02 $^{abc}$ | 40.24 $^{de}$ | 23.14 $^{ab}$ | 13.1 $^{de}$ | 257.08 $^{ab}$ | 145.5 $^{de}$ | 28.28 $^a$ | 19.53 $^a$ |
| SEM | | | 0.04 | 0.17 | 0.39 | 0.16 | 0.33 | 1.81 | 3.69 | 0.32 | 0.28 |
| P Value | | | 0.0498 | 0.0125 | 0.0420 | 0.0089 | 0.04 | 0.0089 | 0.04 | 0.04 | 0.04 |

N =: number of lambs at each strain; BW: birth weight; W3$_{adj}$: adjusted weight at 3 months (weaning); W6$_{adj}$: adjusted weight at 6 months; Diff1: difference of weights at birth to 3 months; Diff2: difference of weights at 3–6 months; ADWG1; average daily weight gain from birth to 3 months; ADWG2: average daily weight gain from 3 to 6 months; FTH: fat-tail height; FTW: fat-tail width. a,b: The means with the same letter in each column are not significantly different in Duncan's multiple range test at 0.05 level.

backcrossing of Texel Tamlet×Moghani (TTM) rams with Moghani pure ewes. It's worth noting that the Texel lines of Tamlet and Dalzell are associated with MyoMAX and double muscling phenotypes in New Zealand Texel sheep, both of which lead to hyperplasia or an increase in muscle fiber count [11]. Despite the BRMBRM-2, BMMBMM-2, TTMTTM-2, and TDMTDM-2 sheep strains, resulting from inbreeding systems among F1 crossbred lambs, displaying lower birth weights, the Texel sheep strains, including TTMTTM-2 and TDMTDM-2, demonstrated significantly higher $W3_{adj}$ and ADWG1 when compared to other sheep strains ($P < 0.05$, see Table 6). These findings suggest that these inbred Texel sheep strains perform better than other sheep strains in terms of reaching the weaning stage.

Regarding fat-tail measurements, BMMBMM-2, TTMTTM-2, TDMTDM-2, and TTMTDM-2, exhibited distinct tail phenotypes compared to other strains. Moghani pure sheep, on the other hand, had significantly higher FTH in comparison to the other strains ($P < 0.05$, as shown in Table 6). As for FTW, no significant difference was observed between Moghani pure sheep and the strains resulting from paternal backcrossing, including TDMM-2, TTMM-2, BMMM-2, and BRMM-2 (refer to Table 6).

## Association of introgressed genes with growth traits

Comparing the results of different genotypes of *MSTN* mutation (OAR2:g.118150665G>A) and *BMPR1B*/*Booroola* mutation (OAR6:g.29382188A>G) on growth traits, it was observed that homozygous reference and heterozygous genotypes for both genes significantly exhibited higher BW compared to the homozygous variant genotype and Moghani pure sheep ($P < 0.05$, as displayed in Table 7). Furthermore, as shown in Table 7, it is noted that Moghani pure lambs exhibited a slightly higher birth weight than F2 crossbred lambs with the *MSTN*

**Table 7. Predicted means for the genotype of introgressed gene on lamb growth traits between pure Moghani sheep and second generations of crossbred lambs (F2 and BC1).**

| Introgressed gene | Genotype | | N | Lamb growth traits | | | | | | |
|---|---|---|---|---|---|---|---|---|---|---|
| | | | | BW (kg) | $W3_{adj}$ (kg) or weaning | $W6_{adj}$ (kg) | Diff1 (kg) | Diff2 (kg) | ADWG1 (gr) | ADWG2 (gr) |
| *MSTN* | homozygous reference | G/G | 103 | 4.77 a | 25.66 b | 49.65 a | 21.07 b | 24.23 a | 234.11 b | 269.22 a |
| | heterozygous | G/A | 94 | 4.74 a | 25.72 b | 47.04 ab | 21.24 b | 21.32 ab | 236.00 b | 236.89 ab |
| | homozygous variant | A/A | 6 | 3.63 b | 30.40 a | 47.11 ab | 26.36 a | 16.75 bc | 292.90 a | 186.13 bc |
| *Booroola* | homozygous reference | A/A | 139 | 4.58 a | 25.62 b | 49.68 a | 21.06 b | 24.06 a | 234.03 b | 267.35 a |
| | heterozygous | A/G | 158 | 4.68 a | 26.03 b | 50.83 a | 21.44 b | 24.80 a | 238.16 b | 275.53 a |
| | homozygous variant | G/G | 4 | 2.56 c | 23.96 b | 39.67 c | 21.45 b | 15.71 c | 238.40 b | 174.53 c |
| **Pure breed** | Moghani wild type | | 172 | 3.88 b | 27.02 b | 40.23 bc | 23.14 b | 13.10 c | 257.08 b | 145.50 c |
| | SEM | | | 0.04 | 0.17 | 0.39 | 0.16 | 0.33 | 1.81 | 3.69 |
| | *P* Value | | | 0.0031 | 0.0187 | 0.0144 | 0.04 | 0.0144 | 0.04 | 0.0144 |

N =: number of lambs at each mating strategy; BW: birth weight; $W3_{adj}$: adjusted weight at 3 months (weaning); $W6_{adj}$: adjusted weight at 6 months; Diff1: difference of weights at birth to 3 months; Diff2: difference of weights at 3–6 months; ADWG1; average daily weight gain from birth to 3 months; ADWG2: average daily weight gain from 3 to 6 months. a,b: The means with the same letter in each column are not significantly different in Duncan's multiple range test at 0.05 level.

mutation in the homozygous variant genotype (A/A), although the difference did not reach statistical significance. In contrast, Moghani pure lambs were significantly ($P < 0.05$) heavier at birth compared to F2 crossbred lambs with the *Booroola* mutation in the homozygous variant genotype (G/G). This aligns with the findings of Çelikeloglu et al. [19] and Tekerli et al. [20], who conducted studies on the introgression of the *MSTN* mutation into Turkish Ramlıç sheep. Their findings indicated that Texel sheep with the *MSTN* mutation in homozygous form (A/A) exhibited significantly lower body weights in comparison to Ramlıç sheep with the wild-type genotype (G/G), as well as to the first-generation backcrosses (BC1, G/G and G/A) and the second-generation backcrosses (BC2, G/G and G/A) lambs [19, 20]. Moreover, the A allele of the *MSTN* mutation OAR2:g.118150665G>A exerted a non-significant adverse impact on live weight traits in Texel sheep [49], Norwegian White sheep [50], Turkish Ramlıç sheep [19], and New Zealand Romney sheep [51]. Nevertheless, these studies did report a statistically significant positive effect on carcass and meat quality traits.

The significant interactions observed between subject effects, specifically strain × genotype and mating system × genotype ($P < 0.0001$), and suggest that birth weight in lambs is influenced by complex genetic and mating factors. The lower birth weight in homozygous variant genotypes could be attributed to the historical in-breeding and outcrossing practices, indicating that genetic diversity plays a crucial role in determining birth weight outcomes [52, 53]. These findings highlight the importance of managing genetic diversity in breeding programs to improve lamb birth weight outcomes. Interestingly, the *MSTN* mutation in the homozygous variant genotype exhibited significantly higher values for $W3_{adj}$, Diff1, and ADWG1 compared to other genotypes and pure Moghani sheep ($P < 0.05$, as shown in Table 7). Our findings align with other studies that have reported significant effects of the *MSTN* mutation OAR2:g.118150665G>A on the growth rate to weaning in New Zealand Romney [54] and Colored Polish Merino Sheep [55]. These results indicate that the *MSTN* mutation OAR2:g.118150665G>A in the homozygous variant genotype has a significant impact on lamb growth until weaning, which is crucial for sheep fattening programs and ewe reproduction management.

In this study we provided additional confirmation that the *Booroola*/*FecB* does not adversely impact the growth rate, consistent with the findings reported in Garole × Malpura sheep [48]. Even in the homozygous variant state, no significant differences were found for Booroola lambs in the genotypes of homozygous reference, heterozygous, and homozygous variant when compared to Moghani pure sheep for the traits of $W3_{adj}$, Diff1, and ADWG1 ($P > 0.05$, as shown in Table 7). We did not explore the impact of the *BMPR1B*/*Booroola* mutation on the prolificacy of crossbred lambs. Despite our thorough examination, prolificacy assessment awaits more data in future investigations. Nonetheless, previous studies have suggested that possessing one copy of the *Booroola* mutation (OAR_v.3.1; Chr 6: g.29382188A/G) leads to an increase in ovulation rate by 1.65 ova and in litter size by 0.9 lambs per lambing. Additionally, ewes that are homozygous (OAR_v.3.1; Chr 6: g.29382188G/G) for this mutation are estimated to experience additional increases of 1.65 ova shed and 0.3 lambs born per lambing [8, 56]. Importantly, Gootwine et al. [23] observed that ewes homozygous for this mutation demonstrate detrimental effects on lamb birth weight, post-weaning growth rate, and mature body weight. It has been underscored that, following the introduction of the *Booroola* mutation, breeding homozygous ewes is not recommended in commercial flocks due to significant lamb losses, despite the exceptionally high prolificacy observed in ewes with this genotype [8].

## Conclusions

This study focused on examining how mating systems influence the performance of Moghani crossbred lambs, which are a mix of the Iranian indigenous Moghani breed and New Zealand

sheep strains, including Texel and Booroola sheep. Among the mating systems employed in the development of second-generation crossbred lambs, both in-breeding and outcrossing consistently produced offspring with lean tails and woolly fleeces, emphasizing the heritability of these desirable traits. However, the examination of backcrossing progeny revealed a spectrum of tail phenotypes, ranging from short to large fat-tails, along with variations in fleece texture. We demonstrated that the utilization of paternal backcrossing emerges as a pivotal strategy for improving the growth rate and overall genetic potential of the Moghani pure breed. The deliberate choice to introgress major genes of *MSTN* and *Booroola* through this method underscores a commitment to precision and efficiency in breeding practices. The observed higher values for growth-related parameters in the *MSTN* mutation homozygous variant genotype indicate a lasting impact on lamb growth until weaning. This finding has significant implications for sheep fattening programs and ewe reproduction management within the Moghani sheep breed. By leveraging the advantages inherent in paternal backcrossing, specifically Texel Tamlet ram strains carrying the *MSTN* mutation, breeders can strategically enhance desirable traits within the Moghani breed, contributing to its resilience, adaptability, and productivity. The study emphasizes the importance of managing genetic diversity and considering both genetic markers and mating systems in sheep breeding programs.

## Supporting information

**S1 Fig. Original image depicting PCR-RFLP band patterns migrated on a 3.5% agarose gel through electrophoresis.**
(PDF)

## Acknowledgments

The authors express their heartfelt appreciation to Jovain Agricultural & Industrial Company, Khorasan Razavi, Jovain, for supplying valuable data. Gratitude is extended to Mr. Hamid Jafar-Abadi, Mr. Mohammad Reza Ghale-Noei, Mr. Valiollah Annabestani, and the diligent animal husbandry team for their collaborative contributions to data collection and blood sampling.

## Author Contributions

**Conceptualization:** Reza Talebi, Mohsen Mardi, Mehrshad Zeinalabedini, Mohammad Reza Ghaffari.

**Data curation:** Reza Talebi, Stéphane Fabre.

**Formal analysis:** Reza Talebi.

**Funding acquisition:** Reza Talebi, Mohsen Mardi, Mehrshad Zeinalabedini, Mohammad Reza Ghaffari.

**Investigation:** Reza Talebi, Mohammad Reza Ghaffari.

**Methodology:** Reza Talebi, Mehrbano Kazemi Alamouti.

**Project administration:** Reza Talebi, Mohsen Mardi, Mehrshad Zeinalabedini, Mohammad Reza Ghaffari.

**Resources:** Reza Talebi.

**Software:** Reza Talebi.

**Supervision:** Reza Talebi, Mohammad Reza Ghaffari.

**Validation:** Reza Talebi, Stéphane Fabre.

**Visualization:** Reza Talebi.

**Writing – original draft:** Reza Talebi.

**Writing – review & editing:** Reza Talebi, Mohsen Mardi, Mehrshad Zeinalabedini, Mehrbano Kazemi Alamouti, Stéphane Fabre, Mohammad Reza Ghaffari.

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
