## [Decision Letter · Decision Letter 0]

29 Feb 2024

PONE-D-23-43880Assessing the performance of Moghani crossbred lambs derived from different mating systems with Texel and Booroola sheepPLOS ONE

Dear Dr. Talebi,

Thank you for submitting your manuscript to PLOS ONE. After careful consideration, we feel that it has merit but does not fully meet PLOS ONE’s publication criteria as it currently stands. Therefore, we invite you to submit a revised version of the manuscript that addresses the points raised during the review process.

Dear Sir,

Please attend the reviewer's suggestions to will publish the paper.

Att

Julio Souza

We look forward to receiving your revised manuscript.

Kind regards,

Julio Cesar de Souza, Ph.D.

Academic Editor

PLOS ONE

Journal Requirements:

Additional Editor Comments:

Dear Authors,

Please attend the reviewer's suggestions to will publish the paper.

Att

Julio Souza

Reviewers' comments:

Reviewer's Responses to Questions

**Comments to the Author**

1. Is the manuscript technically sound, and do the data support the conclusions?

Reviewer #1: Partly

Reviewer #2: Yes

Reviewer #3: Yes

2. Has the statistical analysis been performed appropriately and rigorously? 

Reviewer #1: Yes

Reviewer #2: Yes

Reviewer #3: Yes

3. Have the authors made all data underlying the findings in their manuscript fully available?

Reviewer #1: Yes

Reviewer #2: Yes

Reviewer #3: No

4. Is the manuscript presented in an intelligible fashion and written in standard English?

Reviewer #1: Yes

Reviewer #2: Yes

Reviewer #3: Yes

5. Review Comments to the Author

Reviewer #1: - Introduction is very long

- The means with the same letter in each part of each (column not row) were not significantly different in Duncan’s multiple range test at 0.05 level .

- Others are within the attached manuscript

Reviewer #2: Manuscript Number: PONE-D-23-43880

Manuscript Title: Assessing the performance of Moghani crossbred lambs derived from different mating systems with Texel and Booroola sheep

General Comment: The paper is well presented.

Title: The title is adequate for the content, informative, concise, and clear

Abstract: The abstract effectively summarises the manuscript. However, the authors are requested to add a conclusion following the results and objectives and to avoid subjective language.

Introduction: The introduction provides good, generalised background information on the topic.

Methodology: The study's methods were adequately described and appropriate for the research. The statistical models used in data analysis were correct, and the authors utilized the appropriate statistical package for the analysis.

Results: The results are presented clearly and concisely. However, the interpretation of some results is incorrect in lines 377-378, 382-383, and 496-497. The Tables and Figures are well-organized.

References: The references are appropriate, recent, adequate and cover the works sufficiently. However, the name of the journal is sometimes a full name and sometimes a short name. The authors requested to be consistent.

The numbers below correspond to the line numbers where corrections need to be carried out:

377 – 378: “While females exhibited a slightly improved survival rate compared to males, the observed differences lacked statistical significance” This is incorrect; since there was no significant difference, that means the survival rate of females is similar to males.

382 – 383: Please note that the statement "Intriguingly, F2 crossbred lambs carrying the Booroola gene notably enhanced survivability in comparison to lambs carrying the MSTN gene" is incorrect. There is no significant difference in the survivability rate between lambs carrying the Booroola gene and those carrying the MSTN gene, as shown in Table 2. Therefore, kindly correct this information in both the results and the abstract (lines 34-35).

496 – 497: According to Table 7, it is not true that Moghani pure lambs born heavier than F2 crossbred lambs with the MSTN mutation in the homozygous variant genotype (A/A). There is no significant difference in birth weight between the two (both groups have the same superscript letter).

Reviewer #3: The paper “Assessing the performance of Moghani crossbred lambs derived from different mating systems with Texel and Booroola sheep” presents the results of an ongoing project aimed at evaluating the introgression of Booroola/FecB gene and the 25 myostatin (MSTN) gene into purebred Moghani sheep in the second generation.

The paper is weel-written and presents very interesting results. The introduction characterizes the purpose of the work very well. In the methodology as well results there are many unnecessary topics such as blood collection. In the Results and Discussion, there is no need to repeat the justification and objective of the work. Figures with photos should be improved, as some animals are not to scale. Figures should follow the sequence of text citation and adhere to journal norms. The conclusion is very well-grounded.

6. PLOS authors have the option to publish the peer review history of their article (what does this mean?). If published, this will include your full peer review and any attached files.

Reviewer #1: No

Reviewer #2: No

Reviewer #3: No

---

## [Author Response · Author response to Decision Letter 0]

18 Mar 2024

Response to Reviewers

Reviewer #1: - Introduction is very long

Response: Thank you for the reviewer's comments. we have enhanced the introduction by eliminating surplus and generic information.

Reviewer #1: - The means with the same letter in each part of each (column not row) were not significantly different in Duncan’s multiple range test at 0.05 level .

Response: we appreciate from the reviewer as kindly explained. We have corrected text based on the reviewer explanation (Lines 34-36, Lines 346-349, and Lines 352-354). 

Reviewer #1: - Others are within the attached manuscript

Comment [A1]: Very long

Response: we have refined the introduction in the revised version of the manuscript by removing excess and generic information.

Comment [A2]: Rephrase 

Response: It has been done in revised version of manuscript (lines 67-71). Now it reads " The myostatin (MSTN g+6223G>A) gene located in OAR2 region plays a key role in double muscling across different sheep breeds [11]. These breeds encompass New Zealand Texel [12,13], Australian Texel [14], Belgian Texel [15,16], Norwegian White Sheep [17], and the commercially relevant Charollais sheep [18] "

Comment [A3]: ??????????????

Response: these formula used for correcting the records based on the age of each lamb from birth to weaning and six month age. The formula more clarified in revised version of manuscript (Lines 194-210).

Comment [A4]: Date.W3.BW 

Response: In the revised version of manuscript, we have provided clarity on the formula (located at Lines 194-210 in the revised version of the manuscript) by incorporating appropriate abbreviations for the recorded traits. Additionally, we have revised the abbreviations for each assessed trait throughout the entire manuscript.

Comment [A5]: Sampling time

Response: we would like to clarify that blood samples were obtained from lambs aged between two to three months after birth (Lines 212 in the revised version of the manuscript).

Comment [A6]: details

Response: Thank you for the question. The extraction of genomic DNA from whole blood samples adhered to the procedure outlined by Talebi et al. 2021, as detailed in our prior work. Additionally, the PCR program, RFLP genotyping, and primer information for both the BMPR1B mutation OAR6:g.29382188A>G and the MSTN mutation OAR2:g.118150665G>A were conducted following the methodologies specified in our prior work (Talebi et al. 2023). 

Talebi R, Seighalani R, Qanbari S. A handmade DNA extraction kit using laundry powder; insights on simplicity, cost-efficiency, rapidity, safety and the quality of purified DNA. Anim Biotechnol. 2021;32: 388–394. doi:10.1080/10495398.2019.1684933

Talebi R, Ghaffari MR, Fabre S, Mardi M, Kazemi Alamouti M. Comparison of the growth performance between pure Moghani sheep and crosses with Texel or Booroola sheep carrying major genes contributing to muscularity and prolificacy. Anim Biotechnol. 2023;34: 3495–3506. doi:10.1080/10495398.2023.2165933 

Comment [A7]: ????????

Response: Thank you for the comment. We have corrected the statistical GLM model in revised version of manuscript (Line 225). 

Comment [A8]: delate

Response: Table 1 offers a detailed and comprehensive depiction of coat color patterns, directly facilitating the interpretation of data, bolstering discussions, and significantly enhancing the scientific merit of our manuscript. Consequently, retaining it within the text provides greater informational value to the reader

Comment [A9]: The significant differences. Where is Duncan test 

Response: The descriptive traits presented in Table 2 were analyzed using the chi-square test (X2). The focus of this analysis was to compare the survival rates among different descriptive traits, such as mating systems (in-breeding, outcrossing, paternal backcross, maternal backcross), sex (male and female) and… . The choice of the chi-square test was motivated by our objective to assess whether the observed survival rates significantly deviate from the expected rates within each descriptive trait category. For example, we anticipated equal survival rates across mating systems, sexes or .. but observed variations. The chi-square test was employed to compare observed and expected results, with the resulting p-value indicating the significance level at 5%. To conduct this analysis, we utilized the PROC FREQ procedure in SAS (lines 217-219 of revised version of manuscript), as it is well-suited for frequency tabulations and chi-square tests. 

Comment [A10]: row 

Response: It was corrected based on the reviewer’s comment.

Comment [A11]: Why the authors used Duncan test? This are two groups 

Response: Thank you for your insightful question. In addressing the comparison between two groups (sex: male and female), we initially conducted analyses using both the Duncan test and the LSD (t-test). Upon careful examination, the results obtained from the Duncan test and LSD for the two levels of sex were found to be highly similar. In light of this consistency, we made a deliberate choice to present the results of the Duncan test in Table 4 for the sake of simplicity and ease of interpretation. It is important to note that, in scenarios involving two groups, the Duncan test tends to produce results analogous to the LSD.

Comment [A12]: row 

Response: It was corrected based on the reviewer’s comment.

Comment [A13]: row??????????????????? 

Response: It was corrected based on the reviewer’s comment.

Comment [A14]: row or column 

Response: It was corrected based on the reviewer’s comment.

Reviewer #2: Manuscript Number: PONE-D-23-43880

Manuscript Title: Assessing the performance of Moghani crossbred lambs derived from different mating systems with Texel and Booroola sheep

General Comment: The paper is well presented.

Title: The title is adequate for the content, informative, concise, and clear

Abstract: The abstract effectively summarises the manuscript. However, the authors are requested to add a conclusion following the results and objectives and to avoid subjective language.

Response: We sincerely appreciate the reviewer for his/her thoughtful comments on our manuscript. In response to the reviewer's valuable suggestion, we have included a conclusion after the results and objectives in the abstract of revised version of manuscript (Lines 41-45).

Reviewer #2: Introduction: The introduction provides good, generalised background information on the topic.

Methodology: The study's methods were adequately described and appropriate for the research. The statistical models used in data analysis were correct, and the authors utilized the appropriate statistical package for the analysis.

Results: The results are presented clearly and concisely. However, the interpretation of some results is incorrect in lines 377-378, 382-383, and 496-497. The Tables and Figures are well-organized.

References: The references are appropriate, recent, adequate and cover the works sufficiently. However, the name of the journal is sometimes a full name and sometimes a short name. The authors requested to be consistent.

Response: We value the reviewer's attention to detail regarding the referencing style. In the revised manuscript, we have ensured that all journal names are appropriately abbreviated to match the referencing format of the PLOS ONE journal.

Reviewer #2: The numbers below correspond to the line numbers where corrections need to be carried out:

377 – 378: “While females exhibited a slightly improved survival rate compared to males, the observed differences lacked statistical significance” This is incorrect; since there was no significant difference, that means the survival rate of females is similar to males.

Response: Thank you for the reviewer's valuable comments. As suggested, we have made corrections in the revised version of the manuscript (lines 346-349). It now reads: Further, females displayed a slightly improved survival rate compared to males, the observed differences did not reach statistical significance (refer to Table 2, P > 0.05). Consequently, it can be inferred that the survival rates of females and males are similar. 

Reviewer #2: 382 – 383: Please note that the statement "Intriguingly, F2 crossbred lambs carrying the Booroola gene notably enhanced survivability in comparison to lambs carrying the MSTN gene" is incorrect. There is no significant difference in the survivability rate between lambs carrying the Booroola gene and those carrying the MSTN gene, as shown in Table 2. Therefore, kindly correct this information in both the results and the abstract (lines 34-35).

Response: We sincerely appreciate the reviewer's thoughtful comments. In response to the feedback, we have made corrections in the revised version of the manuscript. It now states, "Intriguingly, F2 crossbred lambs carrying the Booroola gene had no significant difference in survivability compared to lambs carrying the MSTN gene, as indicated in Table 2." (Lines 352-354 in the revised version of the manuscript). Furthermore, this has been rectified in the abstract of the revised version. Now it reads "The F2 crossbred lambs carrying the Booroola gene did not show a statistically significant difference in survivability compared to those carrying the MSTN gene, implying the Booroola prolificacy gene had no significant impact on survival outcomes" (Lines 34-36).

Reviewer #2: 496 – 497: According to Table 7, it is not true that Moghani pure lambs born heavier than F2 crossbred lambs with the MSTN mutation in the homozygous variant genotype (A/A). There is no significant difference in birth weight between the two (both groups have the same superscript letter). 

Response: We sincerely appreciate the thoughtful comments from the reviewer. In response, we have made corrections in the revised version of the manuscript (Lines 465-467). The revised statement now reads: it is noted that Moghani pure lambs exhibited a slightly higher birth weight than F2 crossbred lambs with the MSTN mutation in the homozygous variant genotype (A/A), although the difference did not reach statistical significance.

Reviewer #3: The paper “Assessing the performance of Moghani crossbred lambs derived from different mating systems with Texel and Booroola sheep” presents the results of an ongoing project aimed at evaluating the introgression of Booroola/FecB gene and the 25 myostatin (MSTN) gene into purebred Moghani sheep in the second generation.

The paper is weel-written and presents very interesting results. The introduction characterizes the purpose of the work very well. In the methodology as well results there are many unnecessary topics such as blood collection.

Response: We greatly appreciate the reviewer's positive remarks on our manuscript. We acknowledge the valuable suggestions provided by the reviewer regarding extraneous topics, such as blood collection. In the revised version of the manuscript, we have eliminated the "Blood sample collection" section and seamlessly integrated it into the "Extraction of DNA and genotyping through PCR-RFLP method" section.

Reviewer #3: In the Results and Discussion, there is no need to repeat the justification and objective of the work. 

Response: It was done as kindly suggested by the reviewer. 

Reviewer #3: Figures with photos should be improved, as some animals are not to scale. Figures should follow the sequence of text citation and adhere to journal norms. The conclusion is very well-grounded.

Response: Thank you for your valuable feedback. In revised version of manuscript we addressed this issue by enhancing the visual representation to ensure accurate scaling of all animals. Additionally, we reorganized the figures to align seamlessly with the sequence of text citation and adhere to the prescribed journal norms.

---

## [Editor Report · Decision Letter 1]

20 Mar 2024

Assessing the performance of Moghani crossbred lambs derived from different mating systems with Texel and Booroola sheep

PONE-D-23-43880R1

Dear Dr. Talebi,

We’re pleased to inform you that your manuscript has been judged scientifically suitable for publication and will be formally accepted for publication once it meets all outstanding technical requirements.

Kind regards,

Julio Cesar de Souza, Ph.D.

Academic Editor

PLOS ONE
---

## [Editor Report · Acceptance letter]

26 Mar 2024

PONE-D-23-43880R1 

PLOS ONE

Dear Dr. Talebi, 

I'm pleased to inform you that your manuscript has been deemed suitable for publication in PLOS ONE. Congratulations! Your manuscript is now being handed over to our production team.

Kind regards, 

on behalf of

Dr. Julio Cesar de Souza 

Academic Editor

PLOS ONE